# A Maximum Entropy Model of Bounded Rational Decision-Making with Prior Beliefs and Market Feedback

**DOI:** 10.3390/e23060669

**Published:** 2021-05-26

**Authors:** Benjamin Patrick Evans, Mikhail Prokopenko

**Affiliations:** Centre for Complex Systems, The University of Sydney, Sydney, NSW 2006, Australia; mikhail.prokopenko@sydney.edu.au

**Keywords:** decision-making, bounded rationality, complexity economics, information-theory, maximum entropy principle, quantal response statistical equilibrium, D91, G41, D83, C61, C60, C50

## Abstract

Bounded rationality is an important consideration stemming from the fact that agents often have limits on their processing abilities, making the assumption of perfect rationality inapplicable to many real tasks. We propose an information-theoretic approach to the inference of agent decisions under Smithian competition. The model explicitly captures the boundedness of agents (limited in their information-processing capacity) as the cost of information acquisition for expanding their prior beliefs. The expansion is measured as the Kullblack–Leibler divergence between posterior decisions and prior beliefs. When information acquisition is free, the homo economicus agent is recovered, while in cases when information acquisition becomes costly, agents instead revert to their prior beliefs. The maximum entropy principle is used to infer least biased decisions based upon the notion of Smithian competition formalised within the Quantal Response Statistical Equilibrium framework. The incorporation of prior beliefs into such a framework allowed us to systematically explore the effects of prior beliefs on decision-making in the presence of market feedback, as well as importantly adding a temporal interpretation to the framework. We verified the proposed model using Australian housing market data, showing how the incorporation of prior knowledge alters the resulting agent decisions. Specifically, it allowed for the separation of past beliefs and utility maximisation behaviour of the agent as well as the analysis into the evolution of agent beliefs.

## 1. Introduction

Economic agents are often faced with partial information and make decisions under pressure, yet many canonical economic models assume perfect information and perfect rationality. To address these challenges, Simon [1] introduced bounded rationality as an alternate attribute of decision-making. Bounded rationality aims to represent partial access to information, with possible acquisition costs, and limited computational cognitive processing abilities of the decision-making agents.

Information theory offers several natural advantages in capturing bounded rationality, interpreting the economic information as the source data to be delivered to the agent (receiver) through a noisy communication channel (where the level of noise is related to the “boundedness” of the agent). This representation has spurred the creation of information-theoretic approaches to economics, such as Rational Inattention (R.I.) [2], and more recently, the application of R.I. to discrete choice [3]. Another approach represents decision-making as a thermodynamic process over state changes and employs the energy-minimisation principle to derive suitable decisions [4].

These approaches have shown how one can incorporate a priori knowledge into decision-making, but place no consideration to inferring these decisions based on observed macroeconomic outcomes (e.g., a distribution of profit rates within a financial market) and market feedback loops. Independently, another recent information-theoretic framework, Quantal Response Statistical Equilibrium (QRSE) [5], was developed aiming to infer least biased (i.e., “maximally noncommittal with regard to missing information” [6]) decisions through the maximum entropy principle, given only the macroeconomic outcomes (e.g., when the choice data is unobserved). However, the ways to incorporate prior knowledge into such a system remain mostly unexplored.

In this work, we provide a unification of these approaches, showing how to incorporate prior beliefs into QRSE in a generic way. In doing so, we provide a least biased inference of decision-making, given an agent’s prior belief. Specifically, we show how the incorporation of prior beliefs affects the agent’s resulting decisions when their individual choices are unobserved (as is common in many real-world economic settings). The proposed information-theoretic approach achieves this by considering a cost of information acquisition (measured as the Kullback-Leibler divergence), where this cost controls deviations from an agent’s prior knowledge on a discrete choice set. When the cost of information acquisition is prohibitively high (i.e., when an agent is faced with limitations through time, cognition, cost, or other constraints), the agent falls back to their prior beliefs. When information acquisition is free, the agent becomes a perfect utility maximiser. The cost of information acquisition therefore measures the boundedness of the agent’s decision-making.

The proposed approach is general, allowing the incorporation of any form of prior belief, while separating the agents’ current expectations from their built-up beliefs. In particular, we show how incorporating prior beliefs into the QRSE framework allows for modelling decisions in a rolling way, when previous decisions “roll” into becoming the latest beliefs. Furthermore, we place the original QRSE in the context of related formalisms, and show that it is a special case of the general model proposed in our study, when the prior preferences (beliefs) are assumed to be uniform across the agent choices. Finally, we verify and demonstrate our approach using actual Australian housing market data, in terms of agent buying and selling decisions.

The remainder of the paper is organised as follows. Section 2 provides a background of information-theoretic approaches to economic decision-making, Section 3 describes QRSE and relevant decision-making literature. Section 4 outlines the proposed model, and Section 5 applies the developed model to the Australian housing market. Section 6 presents conclusions.

## 2. Background and Motivation

The use of statistical equilibrium (and more generally, information-theoretic) models remains a relatively new concept in economics [7]. For example, Yakovenko [8] outlines the use of statistical mechanics in economics. Scharfenaker and Semieniuk [9] detail the applicability of maximum entropy for economic inference, Scharfenaker and Yang [10] give an overview of maximum entropy and statistical mechanics in economics outlining the benefits of utilising the maximum entropy principle for rational inference, and Wolpert et al. [11] outline the use of maximum entropy for deriving equilibria with bounded rational players in game theory. Earlier, Dragulescu and Yakovenko [12] showed how in a closed economic system, the probability distribution of money should follow the Boltzmann-Gibbs law [13]. Foley [14] discusses Rational expectations and boundedly rational behaviour in economics. Harré [15] gives an overview of information-theoretic decision-theory and applications in economics, and Foley [16] analyses information-theory and results on economic behaviour.

Ömer [17] provides a comparison of “conventional” economic models and newly proposed ideas from complex systems such as maximum entropy methods and Agent-based models (ABM), which deviate from the assumption of homo economicus—a perfectly rational representative agent. Yang and Carro [18] discuss how a combination of agent-based modelling and maximum entropy models can be complementary, leveraging the analytical rigour of maximum entropy methods and the relative richness of agent-based modelling.

One of the key developments in this area is Quantal Response Statistical Equilibrium (QRSE) proposed by Scharfenaker and Foley [5]. This approach enabled applications of the maximum entropy method [6,19,20] to a broad class of economic decision-making. The QRSE model was further explored in [21], arguing that “any system constrained by negative feedbacks and boundedly rational individuals will tend to generate outcomes of the QRSE form”. The QRSE approach is detailed in Section 3.1.

Ömer [22,23,24] applies QRSE to housing markets (which we also use as a validating example), modelling the change in the U.S. house price indices over several distinct periods, and explaining dynamics of growth and dips. Yang [25] applies QRSE to a technological change, modelling the adoption of new technology for various countries over multiple years and successfully recovering the macroeconomic distribution of rates of cost reduction. Wiener [26,27,28] applies QRSE to labour markets, modelling the competition between groups of workers (such as native and foreign-born workers in the U.S.), and capturing the distribution of weekly wages. Blackwell [29] provides a simplified QRSE for understanding the behavioural foundations. Blackwell further extends this in [30], introducing an alternate explanation for skew, which arises due to the agents having different buy (enter) and sell (exit) preferences. Scharfenaker [31] introduces Log-QRSE for income distribution, and importantly, (briefly) mentions informational costs as a possible cause for asymmetries in QRSE. This is captured by measuring utility *U* as a sum U[a,x]+C(a|x), allowing for higher costs (*C*) of entrance or exit into a market, where *a* is an action and *x* is a rate. Such a separation allows for an “alternative interpretation of unfulfilled expectations”.

These developments show the usefulness of maximum entropy methods, where we have placed particular focus on QRSE, for inferring decisions from only macro-level economic data. However, these approaches do not consider the contribution of a priori knowledge to the resulting decision-making process. The key objective of our study is to generalise the QRSE framework by the introduction of the prior beliefs, as well as the information acquisition costs as a measure of deviation from such priors.

## 3. Underlying Concepts

Two main concepts form the basis for the proposed model. The first is the QRSE approach developed by [5], and the second is a thermodynamics-based concept of decision-making derived from minimising negative free energy, proposed by [4].

### 3.1. QRSE

The QRSE framework aims to explain macroeconomic regularities as arising from social interactions between agents. There are two key assumptions stemming from the idea of Smithian competition: Agents observe and respond to macroeconomic outcomes, and agent actions affect the macroeconomic outcome, i.e., a feedback loop is assumed. It is this feedback that is deemed to cause the macroeconomic outcome to have a distribution that stabilises around an average value. Given only the macroeconomic outcome, QRSE infers the least biased distribution of decisions, which result in the observed macroeconomic distribution using the principle of maximum entropy. This makes QRSE particularly useful for inferring decisions when the individual decision level data is unobserved. In the following section, we outline the key notions behind QRSE [5].

#### 3.1.1. Deriving Decisions

Agents are assumed to respond (i.e., make decisions) based on the macroeconomic outcome, for example, based on profit rates *x*. This is captured by the agents’ utility *U*. However, agents are assumed to act in a boundedly rational way, such that they may not always choose the option with the highest *U*, for example, if it becomes impractical to consider all outcomes. That is, agents are attempting to maximise their expected utility, subject to an entropy constraint capturing the uncertainty: (1)$$\max\sum_{a\in A}f\left[a|x\right]U\left[a,x\right]$$
(2)$$\begin{array}{cc} \mathrm{subject}\;\mathrm{to}\sum_{a\in A}f\left[a|x\right] & =1\\ -\sum_{a\in A}f\left[a|x\right]\log f\left[a|x\right] & \geq H_{min} \end{array}$$
where f[a|x] represents the probability of an agent choosing action *a* if rate *x* is observed. The first constraint ensures the probabilities sum to 1, while the second is a constraint on the minimum entropy. The minimum entropy constraint implies a level of boundedness such that there is some limit to the agents’ processing abilities, which allows QRSE to deviate from perfect rationality.

Lagrange multipliers can be used to turn the constrained optimization problem of Equation (Equation 2) into an unconstrained one, which forms the following Lagrangian function:(3)$$L=-\sum_{a\in A}f\left[a|x\right]U\left[a,x\right]-\lambda \left(\sum_{a\in A}f\left[a|x\right]-1\right)+T\left(-\sum_{a\in A}f\left[a|x\right]\log f\left[a|x\right]-H_{min}\right)$$
taking the first order conditions of Equation (Equation 3), and solving for f[a|x] yields:(4)$$f\left[a|x\right]=\frac{1}{Z}e^{\frac{U[a,x]}{T}}$$
representing a choice of a mixed strategy by maximising the expected utility subject to an entropy constraint. This problem is dual to maximising entropy of the mixed strategy, subject to a constraint on the expected utility as detailed in Section A.1.

#### 3.1.2. Deriving Statistical Equilibrium

From Section 3.1.1 we have a derivation for a decision function, where agents maximise expected utility subject to an entropy constraint introducing bounds in the agents processing abilities. In order to infer the statistical equilibrium based on observed macroeconomic outcomes, the joint probability f[a,x] must be computed.

The joint distribution captures the resulting statistical equilibrium which arises from the individual agent decisions. While there are many potential joint distributions, using the principle of maximum entropy allows for inference of the least biased distribution. From an observer perspective, maximising the entropy of the model accounts for model uncertainty, by providing the maximally noncommittal joint distribution. To compute this, Scharfenaker and Foley [5] maximise the joint entropy with respect to the marginal probabilities (since individual action data is not available), by decomposing the joint entropy into a sum of the marginal entropy and the (average) conditional entropy.

The solution for f[a|x], given by Equation (Equation 4), can be used to compute the joint probability f[a,x], as long as marginal f[x] is determined (since f[a,x]=f[a|x]f[x]). In order to derive f[x], the approach considers the state dependant conditional entropy, represented as
(5)$$H\left[A|x\right]=-\sum_{a\in A}f\left[a|x\right]\log f\left[a|x\right]$$

Scharfenaker and Foley [5] then use the principle of maximum entropy to find the distribution of f[x] which maximises
(6)$$\max_{f[x]\geq 0}H=-\int_{x}f\left[x\right]\log f\left[x\right]dx+\int_{x}f\left[x\right]H\left[A|x\right]dx$$
(7)$$\begin{array}{cc} \mathrm{subject}\;\mathrm{to}\;\int_{x}f\left[x\right]dx & =1\\ \int_{x}f\left[x\right]xdx & =\xi \end{array}$$

The first constraint ensures the probabilities sum to 1, and the second constraint applies to the mean outcome (with ξ being the mean from the actual observed data $\bar{f}\left[x\right]$). Importantly, there is also an additional constraint which models Smithian competition [32] in the market. Smithian competition models the feedback structure for competitive markets, for example, entrance into a market tends to lower the profit rates, and exit tends to raise the profit rates. This is captured as the difference between the expected returns conditioned on entrance, and the expected returns conditioned on exiting. This competition constraint can be represented as
(8)$$\begin{array}{cc} \mathrm{subject}\;\mathrm{to}\;\int_{x}f\left[x\right]\left(f\left[a|x\right]-f\left[\bar{a}|x\right]\right)xdx & =\delta \end{array}$$

The combination of the conditional probabilities of Equation (Equation 4), which stipulate that the agents enter and exit based on profit rates, and the competition constraint of Equation (Equation 8) models a negative feedback loop that results in a distribution of the profit rates around an average (ξ).

Again, using the method of Lagrange multipliers, the associated Lagrangian becomes
(9)$$\begin{array}{c} L=-\int_{x}f\left[x\right]\log f\left[x\right]dx+\int_{x}f\left[x\right]H\left[A|x\right]dx-\\ \lambda \left(\int_{x}f\left[x\right]dx-1\right)-\gamma \left(\int_{x}f\left[x\right]xdx-\xi \right)-\rho \left(\int_{x}f\left[x\right]\left(f\left[a|x\right]-f\left[\bar{a}|x\right]\right)xdx-\delta \right) \end{array}$$
where taking the first order conditions of Equation (Equation 9), and solving for f[x] yields
(10)$$f\left[x\right]=\frac{1}{Z_{A}}e^{H\left[A|x\right]-\gamma x-\rho x(f\left[a|x\right]-f\left[\bar{a}|x\right])}$$
where $Z_{A}$ is the partition function $Z_{A}=\int_{x}e^{H\left[A|x\right]-\gamma x-\rho x(f\left[a|x\right]-f\left[\bar{a}|x\right])}dx$. Note that in Equation (Equation 9) we use ρ as the Lagrangian multiplier for the competition constraint. Parameter ρ is referred to as β in [5], we have avoided this notation to avoid confusion with the thermodynamic β (inverse temperature) discussed in later sections.

Equations (Equation 4) and (Equation 10) comprise a fully defined joint probability. Crucially, QRSE allows for modelling the resultant statistical equilibrium even when the individual actions are unobserved—by inferring these decisions based on the principle of maximum entropy.

#### 3.1.3. Limitations of Logit Response

In Section 3.1.1 we have seen how the logit response function used for decision-making in QRSE is derived from entropy maximisation. Following the Boltzmann distribution well known in thermodynamics, this logit response has seen extensive use throughout the literature arising in a variety of domains. For example, the logit function is used as sigmoid or softmax in neural networks, logistic regression, and in many applications in economics and game theory [33,34]. However, one important development not yet discussed is the incorporation of prior knowledge into the formation of beliefs. Up until now, we have considered a choice to be the result of expected utility maximisation based on entropy constraints from which the logit models have arisen. However, from psychology [35], behavioural economics [36,37], and Bayesian methods [38,39] we know that the incorporation of a priori information is often an important factor in decision-making. Thus, we explore the incorporation of prior beliefs into agent decisions in more detail in the following section (and the remainder of the paper).

Furthermore, one criticism of the logit response arises from the independence of irrelevant alternatives (IIA) property of multinomial logit models (which would extend to the conditional function used in QRSE in a multi-action case), which states that the ratio between two choice probabilities should not change based on a third irrelevant alternative. Initially, this may seem desirable, however, this can become problematic for correlated outcomes (of which many real examples possess). This criticism has been proved correct in several thought experiment studies, showing violations of the IIA assumption [40]. The classical example is the Red Bus/Blue Bus problem [41,42].

Consider a decision-maker who must choose between a car and a (blue) bus, A={car,bluebus}. The agent is indifferent to taking the car or bus, i.e., p(car)=p(bluebus)=0.5. However, suppose a third option is added, a red bus which is equivalent to the blue bus (in all but colour). The agent is indifferent to the colour of the bus, so when faced with $A_{1}=\left\{\mathrm{blue}\;\mathrm{bus},\mathrm{red}\;\mathrm{bus}\right\}$ the agent would choose p(redbus)=p(bluebus)=0.5. Now suppose the agent is faced with a choice between $A_{2}=\left\{\mathrm{car},\mathrm{blue}\;\mathrm{bus},\mathrm{red}\;\mathrm{bus}\right\}$. As per the IIA property, the ratio $\frac{p(\mathrm{blue}\;\mathrm{bus})}{p(\mathrm{car})}$ (from *A*, $\frac{0.5}{0.5}$) must remain constant. So adding in a third option, the probability of taking any *a* becomes $p\left(a\right)=\frac{1}{3}$ (for all *a*), maintaining $\frac{p(\mathrm{blue}\;\mathrm{bus})}{p(\mathrm{car})}=1$. However, this has reduced the odds of taking the car from 0.5 to 0.33 based on the addition of an irrelevant alternative (i.e., the red bus in which the agent does not care about colour of the bus). In reality, the probability for taking the car should have stayed fixed at p(car)=0.5, and the probability of taking a bus reduced to 0.25 each. This reduction in the probability of p(car) does not make sense for a decision-maker who is indifferent to the colour of the bus and is the basis for the criticism. This may not be immediately relevant for current QRSE models (especially binary ones), but with potential future applications, for example, in portfolio allocation, this could become an important consideration. For example, if adding an additional stock to a portfolio which is similar to an existing stock, it may not be desriable to reduce the likelihood of selecting other (unrelated) stocks.

### 3.2. Thermodynamics of Decision-Making

A thermodynamically inspired model of decision-making which explicitly considers information costs, as well as the incorporation of prior knowledge, is proposed by [4]. The proposed approach can be seen as a generalisation of the logit function, where the typical logit function can be recovered as a special case, but in the more general case manages to avoid the IIA property.

Ortega and Braun [4] represent changing probabilistic states as isothermal transformations. Given some initial state x∈X with initial energy potential $\phi_{0}\left[x\right]$, the probability of being in state x is $p\left[x\right]=\frac{e^{-\beta \phi_{0}\left[x\right]}}{\sum_{x^{\prime }\in X}e^{-\beta \phi_{0}\left[x^{\prime }\right]}}$ (from the Boltzmann distribution). Updating state to f[x] corresponds to adding new potential $\Delta \phi_{0}\left[x\right]$. The transformation requires physical work, given by the free-energy difference ΔF[f]. The free energy difference between the initial and resulting state is then
(11)$$\begin{array}{cc} \Delta F[f] & =F[f]-F[p]\\ \; & =\sum_{x\in X}f\left[x\right]\Delta \phi \left(x\right)+\frac{1}{\beta }\sum_{x\in X}f\left[x\right]\log\left(\frac{f[x]}{p[x]}\right) \end{array}$$
which allows the separation of the prior p[x] and the new potential $\Delta \phi_{0}\left[x\right]$. In economic sense, representing the negative of the new potential as the utility gain, i.e., $U\left(x\right)=-\Delta \phi_{0}\left[x\right]$, allows for reasoning about utility maximisation subject to an informational constraint, given here as the Kullback-Leibler (KL) divergence from the prior distribution [4]. Golan [43] shows how the KL-divergence naturally arises as a generalisation of Shannon entropy (of Equation (Equation 2)) when considering prior information, and Hafner et al. [44] show how various objective functions can be seen as functionally equivalent to minimising a (joint) KL-divergence, even those not directly motivated by the free energy principle. Such analysis makes the KL-divergence a logical and fundamentally grounded measure of information acquisition costs, captured as the divergence from a prior distribution.

Ortega and Stocker [45] then apply this formulation to discrete choice by introducing a choice set *A* (space of actions), which leads to the following negative free energy difference, for a given observation *x*:(12)$$-\Delta F\left[f\left[a|x\right]\right]=\sum_{a\in A}f\left[a|x\right]U\left[a,x\right]-\frac{1}{\beta }\sum_{a\in A}f\left[a|x\right]\log\left(\frac{f[a|x]}{p[a]}\right)$$
where again *a* represents a choice (or action), and *U* the utility for the agent. The first term of Equation (Equation 12) is maximising the expected utility, and the second term is a regularisation on the cost of information acquisition. Again, in this representation, information cost is measured as the KL-divergence from the prior distribution.

Taking the first order conditions of Equation (Equation 12) and solving for f[a|x] yields
(13)$$f\left[a|x\right]=\frac{p\left[a\right]e^{\frac{U[x,a]}{T}}}{\sum_{a^{\prime }\in A}p\left[a^{\prime }\right]e^{\frac{U[a^{\prime },x]}{T}}}$$
where we have moved from inverse temperature β to temperature *T* for notational convenience, i.e., $T=\frac{1}{\beta }$. The key formulation here is the separation of the prior probability *p* from the utility gain (or the new potential from the initial potential). *T* then arises as the Lagrange multiplier for the cost of information acquisition (as opposed to the entropy constraint of QRSE, described in Section 3.1). We emphasise this aspect in later sections.

Revisiting the IIA property, the incorporation of the prior probabilities in Equation (Equation 31) can adjust the choices away from the logit equation, and thus managing to avoid IIA. However, if desired, the free energy model reverts to the typical logit function in the case of uniform priors, and so this property can be recovered. In economic literature, a similar model is given by Rational Inattention (R.I.) by [2]. The relationship between R.I. and the free energy approach of [4,45] is detailed in Appendix C.

## 4. Model

In this section, we propose an information-theoretic model of decision-making with prior beliefs in the presence of Smithian competition and market feedback. Given an agent’s prior beliefs and an observed macroeconomic outcome (such as the distribution of returns), the model can infer the least biased decisions that would result in such returns. Importantly, the incorporation of prior beliefs allows for reasoning about the decision-making of the agent based upon both their prior beliefs and their utility maximisation behaviour.

We develop upon the maximum-entropy model of inference from [5], and the thermodynamic treatment of prior beliefs formalised by [4], as outlined in Section 3.

### 4.1. Maximum Entropy Component

The proposed approach can be seen as a generalisation of QRSE, allowing for the incorporation of heterogeneous prior beliefs based on the free-energy principle. The key element is the information acquisition cost, measured as the KL-divergence which arises from the free-energy principle and has been shown to provide a fundamentally grounded application of Bayesian inference [46]. In order to derive decisions f[a|x] for an action or choice *a* (e.g., buy, hold or sell) given an observed return *x* (e.g., a return on investment), we maximise the expected utility *U* subject to a constraint on the acquisition of information measured as the maximal divergence *d* between the posterior decisions and prior beliefs p[a]. As mentioned, *d* is measured as the KL-divergence, which is the generalised extension of the original (Shannon) entropy constraint [43] introduced in Equation (Equation 2)):(14)$$\begin{array}{c} \max\sum_{a\in A}f\left[a|x\right]U\left[a,x\right]\\ \mathrm{subject}\;\mathrm{to}\sum_{a\in A}f\left[a|x\right]\log\left(\frac{f[a|x]}{p[a]}\right)\leq d\\ \sum_{a\in A}f\left[a|x\right]=1 \end{array}$$

The Lagrangian for Equation (Equation 14) then becomes
(15)$$L=\sum_{a\in A}f\left[a|x\right]U\left[a,x\right]-\lambda \left(\sum_{a\in A}f\left[a|x\right]-1\right)-T\left(\sum_{a\in A}f\left[a|x\right]\log\left(\frac{f[a|x]}{p[a]}\right)-d\right)$$

There are two distinct modelling views on such a formulation [47,48,49,50]. The first assumes that specific constraints are known from the data, for example, a maximal divergence *d* may be specified based on actual observations of agent behaviour. The second view, instead, would consider the Lagrange multiplier *T* to be a free parameter of the model, with the constraint *d* representing an arbitrary maximum value: Thus, this approach would optimise *T* in finding the best fit. In this work, we take the second perspective since underlying decision data is unavailable, and a specific restriction on divergent information costs should not be enforced. In other words, *T* is considered to be a free model parameter corresponding to different information acquisition costs, mapping to different (unknown) cognitive and information-processing limits *d*.

Looking at the final term in Equation (Equation 15), in the case of homogeneous priors, logp[a] is a constant which drops out of the solution, which is equivalent to the optimisation problem of Equation (Equation 3), and thus, recovers the original QRSE model. In the general case, the dependence on log(p[a]) means that *T* instead serves as the Lagrange multiplier for the cost of information acquisition. Taking the first order conditions of Equation (Equation 15) and solving for f[a|x] (as shown in Section A.2) yields
(16)$$f\left[a|x\right]=\frac{1}{Z_{A|x}}p\left[a\right]e^{\frac{U[a,x]}{T}}$$
we see this as a generalisation of the logit function, which allows for the separation of the prior beliefs and the agent’s utility function.

In the more general case, p[a] can be heterogeneous for all *a*. Parameter *T* therefore controls the deviations from the prior (rather than from the base case of uniformity), that is, it controls the cost of information acquisition. Following [4], we observe the following limits
(17)$$\begin{array}{cc} \lim_{T\to \infty }f\left[a|x\right] & =p[a]\\ \lim_{T\to 0,T\geq 0}f\left[a|x\right] & =e^{\frac{U[x,a]}{T}}=\max U\left[x,a\right]\\ \lim_{T\to 0,T<0}f\left[a|x\right] & =e^{\frac{U[x,a]}{T}}=\min U\left[x,a\right] \end{array}$$

In the limit T→∞ (i.e., infinite information acquisition costs), the agent just falls back to their prior beliefs as it becomes impossible to obtain new information. In the limit T→0, the agent becomes a perfect utility maximiser (i.e., if information is free to obtain, the agent could obtain it all and choose the option that best maximises payoff with probability 1). In the T<0 case, we see this corresponds to anti-rationality. For economic decision-making, we can limit temperatures to be non-negative, T≥0, although there are specific cases where such anti-rationality may be useful (e.g., modelling a pessimistic observer or adversarial environments [4]). The relationship between temperature and utility is visualised in Figure 1.

Crucially, large temperatures (costly acquisition) do not revert to the uniform distribution (as in the typical QRSE case, unless the prior is uniform), instead reverting to prior beliefs. This is visualised in Figure 2, and discussed in more detail in Section 4.3.

### 4.2. Feedback Between Observed Outcomes and Actions

Following [5], we use a joint distribution to model the interaction between the economic outcome *x*, and the action of agents *a*.

To recover a joint probability, we need to determine f[x] (since f[a,x]=f[a|x]f[x]) which we do with the maximum entropy principle, as shown in Section 3.1. To do this, we maximise the joint entropy with respect to the marginal probabilities. That is,
(18)$$\begin{array}{c} L=-\int_{x}f\left[x\right]\log f\left[x\right]dx+\int_{x}f\left[x\right]H\left[A|x\right]dx-\lambda \left(\int_{x}f\left[x\right]dx-1\right)\\ -\gamma \left(\int_{x}f\left[x\right]xdx-\xi \right)-\rho \left(\int_{x}f\left[x\right]\;\frac{p\left[a\right]e^{\frac{U[a,x]}{T}}-p\left[\bar{a}\right]e^{\frac{U[\bar{a},x]}{T}}}{Z_{A|x}}xdx-\delta \right) \end{array}$$
with
(19)$$\begin{array}{cc} H[A|x] & =-\sum_{a\in A}f\left[a|x\right]\log f\left[a|x\right]\\ \; & =-\frac{1}{Z_{A|x}}\sum_{a\in A}p\left[a\right]e^{\frac{U[a,x]}{T}}\left(\log p\left[a\right]+\frac{U[a,x]}{T}-\log Z_{A|x}\right) \end{array}$$

An important point to be made here is that H[A|x] still measures (Shannon) entropy. We have seen above how the new definition for f[a|x] uses the KL-divergence as a generalised extension of entropy when incorporating prior information. In Equation (Equation 19), we do not use this divergence for an important reason. In Equation (Equation 14) we are measuring divergence from known prior beliefs, however, now when optimising Equation (Equation 18) we wish to infer decisions from unobserved decision data. This is where the principle of maximum entropy comes into play, i.e., we wish to maximise the entropy of our new choice data (which was derived from KL-divergence of prior beliefs), but we do not wish to perform cross-entropy minimisation as we do not have the true decisions $\bar{f}\left[a|x\right]$. With this in mind, we still utilise the principle of maximum entropy as is done in QRSE for inference to obtain the least biased resulting decisions. This keeps the proposed extensions in the realm of QRSE, but comparisons to the principle of minimum cross-entropy [51,52] could be considered in future work particularly when some target distributions are known directly.

In Equation (Equation 18), ξ is known from the mean of the observed macroeconomic outcome, and so this constraint is used explicitly. This is in contrast to *d* (and δ) which are unknown as outlined in Section 4.1. The important distinction with Equation (Equation 18) is that the f[a|x] functions (and H[A|x]) now use the updated expressions for f[a|x], which incorporate the prior beliefs. Taking the partial derivative of L with respect to f[x], and solving for f[x] gives
(20)$$\begin{array}{cc} f[x] & =\frac{1}{Z_{A}}e^{H\left[A|x\right]-\gamma x-\rho x\left(\frac{p\left[a\right]e^{\frac{U[a,x]}{T}}-p\left[\bar{a}\right]e^{\frac{U[\bar{a},x]}{T}}}{Z_{A|x}}\right)} \end{array}$$

Equation (Equation 20) expresses the information acquisition cost in the form of the Lagrange multiplier *T* (from Equation (Equation 15)), and a competition cost in the form of the multiplier ρ.

As we have a solution for f[a|x] (Equation (Equation 16)) and f[x] (Equation (Equation 20)) in terms of prior beliefs and information acquisition costs, we can then derive all other probability functions using the Bayes rule. That is, we can obtain f[a,x], f[x|a] and f[a] which in turn incorporate these prior beliefs/acquisition costs:(21)$$\begin{array}{cc} f[a,x] & =f[a|x]f[x]\\ \; & =\frac{p\left[a\right]e^{\frac{U[a,x]}{T}+H\left[A|x\right]-\gamma x-\rho x\left(\frac{p\left[a\right]e^{\frac{U[a,x]}{T}}-p\left[\bar{a}\right]e^{\frac{U[\bar{a},x]}{T}}}{Z_{A|x}}\right)}}{Z_{A|x}Z_{A}} \end{array}$$

We can obtain f[a] by marginalising out *x* from the joint distribution:(22)$$\begin{array}{cc} f[a] & =\int_{x}f\left[a,x\right]\\ \; & =\frac{1}{Z_{A}}\int_{x}\frac{1}{Z_{A|x}}p\left[a\right]e^{\frac{U[a,x]}{T}+H\left[A|x\right]-\gamma x-\rho x\left(\frac{p\left[a\right]e^{\frac{U[a,x]}{T}}-p\left[\bar{a}\right]e^{\frac{U[\bar{a},x]}{T}}}{Z_{A|x}}\right)} \end{array}$$

Finally, f[x|a] can then be computed by a direct application of the Bayes rule: f[x|a]=f[a,x]/f[a].

Given only an expected average value ξ (and the usual normalisation constraints), we have derived a joint probability distribution, which maximises the entropy subject to some information acquisition cost *d*, along with a competition cost δ. The resulting distribution free parameters (the Lagrange multipliers) are those which fit most closely to the true underlying distribution of returns. Thus, we have provided a generalisation of QRSE, which is fully compatible with the incorporation of prior beliefs.

### 4.3. Priors and Decisions

The introduced priors affect the conditional probabilities of agent decisions by shifting focus towards these preferred choices. The introduced priors allow the decision-maker to place more focus on particular actions if they have been deemed important a priori.

In Section 3.2 we showed how to separate the initial energy potential and new energy potential for distinguishing prior beliefs and utility functions. It is instructive to interpret these again as potentials, by setting $\alpha_{a}=T\log p\left[a\right]$, which allows us to represent the choice probability as
(23)$$f\left[a|x\right]=\frac{1}{Z_{A|x}}e^{\frac{U\left[x,a\right]+\alpha_{a}}{T}}\;.$$

Equation (Equation 23) shows how α shifts the likelihood based on the prior preferences. An example of these shifts is visualised in Figure 2. This can be interpreted as placing more emphasis on actions deemed useful a priori as *T* increases. The information acquisition cost component *T* then controls the sensitivity between the utility and a priori knowledge, with a high *T* meaning higher dependence on prior information, and low *T* indicating a stronger focus on the utility alone.

The majority of binary QRSE models use a simple linear payoff definition for utility:$$U\left[x,a\right]=x-\mu ,\;\;U\left[x,\bar{a}\right]=-\left(x-\mu \right)\;.$$

With this definition, a tunable shift parameter μ serves as the expected fundamental rate of return. The relationship between μ and the real markets returns ξ (which was used as a constraint in Equation (Equation 7)), serves then as a measure of fulfilled expectations (i.e., if μ = ξ) or unfulfilled expectations (μ≠ξ). This implies a symmetric shift parameter μ. As a specific example, if a=sell and $\bar{a}=\mathrm{buy}$, μ=0.25 means that at x=0.25, buyers and sellers will be equally likely to participate in the market, i.e., f[sell|μ]=f[buy|μ]=0.5. In this sense, μ can be seen as the indifference point. The symmetry arises from the fact that f[buy|x]+f[sell|x]=1. Therefore, in the binary action case, it is possible to find a $\mu^{*}$ with the uniform priors p=[0.5,0.5] such that the decision functions will be equivalent to μ with any arbitrary priors p=[c,1−c], with c∈[0,1]. In this sense, μ can be seen as encapsulating a prior belief.

However, explicit incorporation of prior beliefs on actions is useful here as it helps to separate the agents’ expectations in relation to their prior belief (e.g., a higher μ resulted from needing to change from their past behaviour) and choose the actions for which an agent should emphasise acquiring more information. The introduced prior beliefs are strictly known before any inference is performed, whereas μ is the result of the inference process. The separation of prior beliefs and current expectations is important, as with μ alone this can not capture an agent’s predisposition prior to performing any information processing. In addition, this applies more generally to any arbitrary utility functions (as QRSE is, of course, not limited to the linear shift utility function with μ outlined above), or when any preference is known about decisions a priori.

Consider also the three action case, A={buy, hold, sell}, with the same utility functions as above but with the extra utility for holding being U[x,hold]=0. We can see that it would be desirable if buying and selling no longer required this symmetry. The use of priors can introduce this asymmetry, by providing separate indifference points for buy/hold and sell/hold. Such asymmetry alters the resulting frequency distribution of transactions, and may help to explain various trading patterns [16]. The difference of symmetric and asymmetric buy and sell curves is shown in Figure 3. Figure 3 shows that such functions could be recovered by introducing a secondary shift parameter $\mu_{2}$. Parameter $\mu_{1}$ (the original μ) then becomes the indifference point for buy and hold, and $\mu_{2}$ for sell and hold. This is the method proposed in [30]. Introducing priors into this case again allows for separation of expectation μ, from prior belief and follows the same methodology as outlined above for the binary case. Furthermore, if we set p[hold]=0, we recover the binary case. This highlights that the standard QRSE with binary actions and uniform priors is a special case of the ternary action case with heterogeneous priors.

From this, we can see how introducing priors alters the decision functions by allowing agents to focus on suitable a priori candidate actions. We have also shown how the binary case of a utility function with a shift parameter can be formalised to achieve equivalent results with a uniform prior and altered shift parameter. However, in the multi-action case, the priors allow for asymmetry, and in general, the priors may help with the optimisation process (by providing an alternate initial configuration). This approach also allows for the explicit separation of the two factors affecting an agent’s choice, by distinguishing the contributions of prior beliefs and the utility maximisation.

### 4.4. Rolling Prior Beliefs

The proposed extension is general and allows for the incorporation of any form of prior beliefs, and in this section, we illustrate an example where the priors at time *t* are set as the resulting marginal probabilities from the previous time t−1:$$p_{t}\left[a\right]=f_{t-1}\left[a\right]$$
i.e., the prior belief $p_{t}\left[a\right]$ is set as the previous marginal probability $f_{t-1}\left[a\right]$ for taking action *a* (at t=0, we use a uniform prior). Using the previous marginal probability as a prior introduces an “information-switching” cost, where *T* relates to the divergence from the previous actions, resulting in the following decision function:$$f_{t}\left[a|x\right]=\frac{1}{Z_{A|x}}f_{t-1}\left[a\right]e^{\frac{U[x,a]}{T}}$$

That is, acquiring information on top of the previous knowledge comes at a cost (controlled by *T*). When the cost of information acquisition is high (large *T*), the agent reverts to the previously learnt knowledge (i.e., the marginal probabilities from t−1). In contrast, when *T* is extremely small, the agent is able to acquire new information allowing deviation from their prior knowledge at t−1. In the special case of T=0, information is free, and the agent can become a perfect utility maximiser.

Given the expression for $f_{t}\left[a|x\right]$, we obtain the following solution for $f_{t}\left[x\right]$:$$f_{t}\left[x\right]=\frac{1}{Z_{A}}e^{H\left[A|x\right]-\gamma x-\rho x\left(\frac{f_{t-1}\left[a\right]e^{\frac{U[a,x]}{T}}-f_{t-1}\left[\bar{a}\right]e^{\frac{U[\bar{a},x]}{T}}}{Z_{A|x}}\right)}$$
from which we can derive the joint and other probabilities, as shown in Section 4.1. This is exemplified in Section 5, in which we examine various priors for time-dependent applications.

## 5. Australian Housing Market

To exemplify the model, we use the Greater Sydney house price dataset provided by SIRCA-CoreLogic and utilised in [53,54]. This dataset is outlined in Appendix B. In [54], an agent-based model is used to explain and forecast house price trends and movement patterns as arising from the individual agent’s buy and sell decisions. Furthermore, the ABM implemented bounded rational agents driven by social influences (e.g., fear of missing out) and partial information about submarkets. While the resulting dynamics produced by the ABM accurately match the actual price trends, the decision-making mechanism and the bounded rationality of the agents were not theoretically grounded. In the following section, we aim to explain how the bounded rational behaviour of the agents operating in the housing market can be aligned with the model proposed in this study based on prior beliefs of agents and Smithian competition within the market. With this example, Smithian competition can be seen as agent decisions (buying or selling) affecting returns for an area, and agents decisions also being made based on returns for particular areas, i.e., a feedback loop is assumed in the market.

In particular, we want to explore what role an agent’s prior beliefs play in their resulting decisions. For example, given equivalent configurations (e.g., utility and returns) and different prior knowledge, how would the agent’s behaviour differ? Furthermore, we would like to explore the rationality of the agents, measured in terms of the cost of information acquisition, in order to see how the agents behave. For example, are agents predominantly reliant on past knowledge in times of market growth, resulting in unexpected downturns from mismanaged agent expectations? Alternatively, in deciding if it is a good time to buy or sell, the agents may balance their past knowledge with utility and current returns (i.e., the past knowledge would not be a predominant factor). The proposed model is particularly suited for answering such questions due to the low number of free (and microeconomically) interpretable parameters, as well as the explicit separation of prior beliefs (as opposed to previous QRSE approaches). Our goal is not to infer the “best” prior, but rather to explore and compare dynamics resulting from various priors. In addition, we aim to verify the conjecture that during crises, and periods exhibiting non-linear market dynamics, macroeconomic conditions may become more heterogeneous, and thus, non-uniform priors may outperform uniform ones in such times.

### 5.1. Model

We use our model of binary actions with prior beliefs introduced in Section 4.1, with actions A={buy,sell}. The decision functions are then given by
(24)$$\begin{array}{c} f_{t}\left[\mathrm{buy}|x\right]=\frac{1}{Z_{t,x}}p_{t}\left[\mathrm{buy}\right]e^{\frac{U[x,\mathrm{buy}]}{T}}\\ f_{t}\left[\mathrm{sell}|x\right]=\frac{1}{Z_{t,x}}p_{t}\left[\mathrm{sell}\right]e^{\frac{U[x,\mathrm{sell}]}{T}}\\ Z_{t,x}=f_{t}\left[\mathrm{buy}|x\right]+f_{t}\left[\mathrm{sell}|x\right] \end{array}$$
where we explore a range of $p_{t}$ (prior at time *t*) functions, discussing their effects on decision-making and resulting probability distributions.

#### 5.1.1. Priors

While the proposed approach is capable of incorporating any form of prior belief on the choice set *A*, below we outline several example priors which we explore. In exploring these priors, we highlight differences in resulting agent posterior decisions based on various prior beliefs.

##### Uniform

We begin with a uniform prior. The uniform probability represents the default case of QRSE, where each action has an equally weighted prior. In the binary case, this corresponds to $p_{t}\left[a\right]=0.5$ for all *t* and *a*. This corresponds to an agent who is agnostic to the available actions before observing *U*.

##### Previous

Next we look at a “previous” prior. The previous prior uses the marginal action probabilities from the previous time step as priors to the current timestep. This means at time *t*, $f_{t}\left[a\right]$ plays the role of a posterior probability of making a decision, however, at time t+1$f_{t}\left[a\right]$ now serves as the empirical prior. This is the example introduced in Section 4.4. This corresponds to $p_{t}\left[a\right]=f_{t-1}\left[a\right]$ for t>0, and $p_{t}\left[a\right]=0.5$ for t=0. The previous prior represents an empirical prior where the decision is conditioned on previous market information, where *T* controls the level of influence from the previous market stage (in our case, each year). A high *T* means high influence from the past market state, whereas low *T* means focusing on current market conditions alone (as measured by *U*). In the extreme case of T=∞, a backward looking expectations [55] approach is recovered where decisions are assumed to be a function purely of past decisions, however, in the more general case with T<∞, *U* adjusts the decisions based on the current market state.

##### Mean

We also consider a mean prior. The mean prior uses the average marginal action probability from all previous timesteps. This corresponds to $p_{t}\left[a\right]=\frac{\sum_{t^{\prime }=0}^{t-1}f_{t^{\prime }}\left[a\right]}{t}$, for t>0, and $p_{t}\left[a\right]=0.5$ for t=0. This can be seen as belief evolution, where over time, the previous decisions help build the current prior (modulated by *T*) at each stage.

##### Extreme Priors

As two further examples, we introduce extreme priors (more for visualisation/discussion sake as opposed to being particularly useful). The extreme buy prior corresponds to a strong prior preference for the buy action, $p_{t}\left[\mathrm{buy}\right]=0.99,p_{t}\left[\mathrm{sell}\right]=0.01$, for all *t*. Likewise, the extreme sell case is simply the inverse of the buy case, a strong prior preference for selling, i.e., $p_{t}\left[\mathrm{sell}\right]=0.99,p_{t}\left[\mathrm{buy}\right]=0.01$, for all *t*.

However, the formulations provided above by no means represent an exhaustive set of possible priors. For example, Genewein et al. [56] discuss “optimal” priors, which draws parallels with rate-distortion theory and can be seen as building abstractions of decisions (see Appendix C). Adaptive expectations [57] are discussed in [58,59,60], where priors could be partially adjusted based on some strength term ($\lambda_{E}$), where the strength term adjusts the contribution from some error. For example, an adaptive prior could be represented as $p_{t}=p_{t-1}+\lambda \left(p_{t-1}-{\hat{p}}_{t-1}\right)$, where ${\hat{p}}_{t-1}$ is the actual known likelihood of actions from the previous time period. With our specific housing market data, we do not have $\hat{p}$, i.e., we do not have the true buying and selling likelihoods, but if known, such information could be used to adjust future beliefs, i.e., over time the adaptive priors would adjust decisions based on the previously observed likelihoods (controlled by λ). The proposed approach makes no assumption about the forms of prior beliefs, so the ideas outlined above can be incorporated into the method outlined here by adjusting the definition of $p_{t}$.

### 5.2. Results

We fit the distributions with the various priors outlined in Section 5.1.1 to the actual underlying return data, to estimate how well we are able to capture this distribution and explore the effects that these priors have on the resulting distribution. The results are presented in Table 1, which summarises the likelihood and the percentage of the explained variability (measured as Information Distinguishability (I.D.) [61]) compared to the underlying distribution. We see that there are no large differences in general between the priors in terms of the explained variability. However, the goal here is not to argue for the “best” prior fitting the dataset in terms of the explained variability, but rather to explore differences in the agent behaviour based on the prior knowledge (using the housing dataset as an example). Thus, the resulting fitted distributions f[x], which are visualised in Figure A5, are more interesting. We observe how altering prior beliefs result in different resulting distributions and discuss how the incorporation of prior beliefs allows for a separation of the agents’ utility maximisation behaviour from their previous knowledge. From Figure A5 we can also see how the priors can alter the optimisation process, for example, a good (bad) prior may help (harm) the optimisation by providing alternate initial configurations. The extreme priors can be seen as harmful, for example, in 2012 where the resulting distributions are unable to capture the true underlying distribution. The reason for this is being unable to find suitable *T* to enable appropriate divergence from the extreme prior beliefs. In contrast, well selected priors can help the optimisation process and result in better fitting distributions, such as in 2016 where the decisions resulting from the mean and previous prior fit the true data significantly better than the uniform prior.

The agents’ decision functions f[a|x] are visualised in Figure A7 which makes it clear how each prior adjusts the resulting probability of taking an action (and thus, alters the decisions). From this, we can see different probabilistic behaviours despite having equivalent utility functions and optimisation processes due to varying prior beliefs. For example, with the extreme priors, we observe a clear shift towards the strongly preferred action.

Figure A6 shows the resulting joint distributions f[a,x], combining the results of Figure A5 and Figure A7, since f[a,x]=f[a|x]f[x]. Looking at the second row of each plot in Figure A6, we can see a visual representation of how the joint probabilities adjust over time when using the previous year as the prior belief.

The resulting marginal action probabilities are visualised in Figure 4, where we observe clear market peaks and dips which match the actual returns of Figure 5, aligning with the general trends observed in Figure A1. The priors work on either increasing or decreasing the resulting marginal probabilities. For example, in the extreme sell case we see much higher resulting probabilities for f[sell], likewise in the extreme buying case, we see much higher probabilities for f[buy]. The general peaks/dips remain in both cases. Overall, this shows how the prior belief can influence the resulting marginal probabilities.

Using the previous year’s marginal probability as a prior for the current year has a smoothing effect on the resulting year-to-year marginal probabilities. Comparing the previous prior with the uniform prior in Figure 4, we observe, particularly during 2015–2018, a more defined/well-behaved step-off in f[sell]. This indicates the slowing of returns during these years. At the same time, the uniform priors are more affected by local noise, potentially overfitting to only the current time period, since no consideration can be given to the past behaviour of the market. This results in larger fluctuations in the agent behaviour as they have no concept of market history.

### 5.3. Role of Parameters

One of the benefits of QRSE is the low number of free parameters which results in a relatively interpretable model. There are four free parameters in the typical QRSE distribution: T,μ,ρ and γ, each with a corresponding microeconomic foundation. In this section, we discuss the two main parameters of interest in this work: The decision temperature *T* and agent expectations μ, and the effect that prior beliefs have on the resulting values (and interpretation) of these parameters. We also include discussion on the impact of decisions on resulting outcomes ρ and skewness of the resulting distributions γ in Appendix D, since ρ and γ were less affected by the introduced extensions. There is an additional parameter ξ (shown in Figure 5), which is not a free parameter, representing the mean of the actual returns and serving as a constraint on the mean outcome in Equation (Equation 7).

#### 5.3.1. Decision Temperature

The decision temperature *T* controls the level of rationality and deviations from an agent’s prior beliefs. An extremely high temperature corresponds to high information acquisition cost and results in choosing actions simply based on the prior belief. In contrast, an extremely low temperature corresponds to utility maximisation, and in the case of free information (T=0) a perfect utility maximiser is recovered (i.e., homo economicus). In the housing example used here, *T* relates to the ability of an agent to learn all the required knowledge of the market, i.e. the actual profit rates for various areas. With T=0, the agent has perfect knowledge of the current market profitability. With T>0, this represents some friction with acquiring such information, e.g., it can be difficult to gather all the required information to make an informed choice due to, for example, search costs. From a psychological perspective, *T* can be a measure of the “just-noticeable difference” [62], meaning microeconomically, *T* is related to the ability of an agent to observe quantitative differences in resulting choices. High *T* means the agent is unable to distinguish choices based on *U*, due to high information-processing costs, so instead acts according to their previously learnt knowledge.

Since *T* is related to the prior, we see differences in the resulting values visualised in Figure 6. What can be observed from looking at the general trends of *T* is that it peaks in the years with high average growth (large ξ), such as 2015, as these years correspond to a growing market, and agents require less attention to market conditions, although this depends on the prior used.

Looking at the previous marginal probability as the prior (the orange profile), we observe in the build-up phase to 2015 increasing decision temperatures corresponding to agents acting on these previous beliefs. As these beliefs were also positive (i.e., agents expected favourable returns), these large returns can be explained by the agents continuously expecting this growth. This pattern changed in 2016, when the market “reverses”: Now the agents must focus instead on their current utility since their prior beliefs no longer reflect the current market state. Such market reversals are categorised by low decision temperatures, since using the previous action probabilities now becomes misinformative (in contrast to the “building”/trend-following stages). This indicates an increased focus on agent rationality in times of market reversals. The incorporation of prior beliefs (particularly using the previous priors) is useful as it allows for the discussion to be extended in the temporal sense (as is done here). In other words, we can consider “building” the agent’s beliefs as possible underlying causes for market collapses and relating the rationality of agents to the relative state of the market.

#### 5.3.2. Agent Expectations

In microeconomic terms, parameter μ captures the agent’s expectations. A large μ corresponds to an optimistic agent, who is expecting high returns from the market. In contrast, a low μ corresponds to a pessimistic agent, who is expecting poor returns from the market. As this works to shift the decision functions, there is a relation between the prior and parameter μ, since the prior also works as shifting preferences towards a priori preferred actions as shown in Section 4.3. There is also a relationship between μ and γ (outlined in Section D.2), since γ can help to account for unfulfilled agent expectations by adjusting the skew of the resulting distributions.

Generally, the agent’s beliefs are within the ±2.5% range (expecting between a 2.5% quarterly growth or 2.5% dip), which corresponds to the bulk of the area under the curve in Figure A2. This means that the agent’s expectations develop in accordance with actual market conditions, as can be seen in Figure 7.

The extreme priors result in larger absolute values of μ since larger shifts are needed to offset the (perhaps) poor prior beliefs. This can be seen in 2014 particularly, where the extreme sell prior has μ=10%.

The values of previous prior μ tend to have a larger magnitude than the uniform priors, since as mentioned, these priors can capture build-up of beliefs (and as such some “trend-following” can be captured). For example, the year 2008 saw the lowest average returns ξ, as shown in Figure 5. Using the previous prior, the agents’ expectations correctly match the sign of the actual returns in 2008 (i.e., agents correctly expected a decline in house prices). This results in more pessimistic agents than those using the uniform prior since they can reflect on the market performance from 2007. Likewise, during 2013–2015, the values of previous prior μ become larger than those for the uniform prior, since they are building on the previous years expectations which were all positive. In contrast, the period 2015–2017 saw a steady decline in agents expectations of returns with previous priors, reflecting the overall market state which appeared to be in a downward trend. The previous priors were able to capture this trend. Using the uniform priors, the year 2016 had a higher μ than the market peak of 2015. The reason is that uniform priors are unable to capture the fact that the previous timestep had higher (or lower) returns than the current timestep. In this case, the discussion can not be extended in the temporal sense of “building" on beliefs, and agents may miss such crucial temporal information without the incorporation of prior beliefs. This is evidenced by the significantly lower performance of the uniform prior in 2016 in comparison to the previous prior, as shown in Table 1, highlighting the usefulness of non-uniform (and temporal-based) priors in times of market crises and reversals.

### 5.4. Temporal Effects of Data Granularity on Decisions

In Section 5.2, we have analysed agent decisions over the previous 15 years, where decisions were grouped annually. This level of granularity was chosen to examine different agent behaviour from year to year. However, other levels of grouping can also be explored to give an insight into the impact of noise on the inference process. For example, an extremely granular grouping will likely result in additional noise in the decision-making process, which may or may not be impacted by the incorporation of prior beliefs. Likewise, a low granular grouping can be seen as “pre-smoothed”, which may work in a similar fashion to the incorporation of prior temporal-based beliefs at a higher granularity, which we have seen can smooth the resulting decisions. In this section, we examine the usefulness of prior beliefs in such situations, providing comparisons with alternate data representations.

Two additional levels of granularity are considered, one more granular and one less granular than the annual groupings introduced in Section 5.2. We look at quarterly data, as well as aggregate groupings based on market state. In doing so, we have three levels for categorising agent behaviour: Quarterly, annually, and aggregated market state. This allows us to compare resulting agent decisions across different temporal scales, comparing the differences generated by the incorporation of prior beliefs and various data-level modifications.

The aggregate market state data groups years into “terms”, which correspond with various “stages” of the market. These are growth and crash phases, highlighted as “Pre Crash” (Mid 2006–2007), “Crash” (2008), “Recovery 1” (2009–Mid 2011), “Small Crash” (Mid 2011–Mid 2012), “Recovery 2” (Mid 2012–Mid 2018) and “Recent Crash” (Mid 2018 to 2020). The overall market trends can be visualised in Figure A1 to see market returns for each corresponding “term”.

The resulting decision likelihoods f[A] are presented in Figure 8. In analysing the differences in resulting marginal probabilities between the various granularities, we can observe the impact from data-level modifications, i.e., performing inference on a larger time scale for macroeconomic observations, and how the incorporation of prior information affects such results. In Section 5.2 we have mentioned the previous and mean priors can have a smoothing effect on resulting decisions, in this sense, the lower granularity groupings (the market state based grouping) can also be seen as a smoothed version of the macroeconomic outcomes, i.e. pre-smoothing the data by considering a much larger interval composed of several years for groupings. We see that the incorporation of prior information helps preserve some important information in such settings. Looking at the left-most column of Figure 8 (the uniform priors), we can see the overall “shape” of the peaks and dips in preferences f[a] is lost with aggregate groupings. For example, in the quarterly breakdown, there is a clear preference for selling in the later region in the range 2014–2017, corresponding to the highest growing market, which is labelled as “Recovery 2” in the aggregated version. When considering the “Recovery 2” with uniform priors, such a clear preference is lost, and the “Pre Crash” and “Initial Recovery” have a higher corresponding preference. This is because the agents can not separate past market information from the current market state and act purely based on the current utility. In contrast, with both the mean and the previous prior, such overall trends are preserved across the various granularities since agents can distinguish favourable environments when compared with previous market states (as captured by their prior beliefs). This additional temporal insight provides an important consideration and shows that even with various data-level smoothing or preprocessing (i.e., considering alternate data groupings) the prior information remains useful and highlights various market states and corresponding agent preferences.

A key takeaway from this exploration is that the potential for temporal analysis introduced by the prior beliefs provides additional insights into decision-making. These insights can not be generated by simple data-level modifications. Furthermore, the decision temperature *T* provides a way to modulate market state changes when considering agent decision-making.

## 6. Discussion and Conclusions

Despite many well-founded doubts of perfect rationality in decision-making, agents are often still modelled as perfect utility maximisers. In this paper, we proposed an approach for inference of agent choice based on prior beliefs and market feedback, in which agents may deviate from the assumption of perfect rationality.

The main contribution of this work is a theoretically grounded method for the incorporation of an agent’s prior knowledge in the inference of agent decisions. This is achieved by extending a maximum entropy model of statistical equilibrium (specifically, Quantal Response Statistical Equilibrium, QRSE), and introducing bounds on the agent processing abilities, measured as the KL-divergence from their prior beliefs. The proposed model can be seen as a generalization of QRSE, where prior preferences across an action set do not necessarily have to be uniform. However, when uniform prior preferences are assumed, the typical QRSE model is recovered. The result is an approach that can successfully infer least biased agent choices, and produce a distribution of outcomes matching that of the actual observed macroeconomic outcomes when individual choice level data is unobserved.

In the proposed approach, the agent rationality can vary from acting purely on prior beliefs, to perfect utility maximisation behaviour, by altering the decision temperature. Low decision temperatures correspond to rational actors, while high decision temperatures represent a high cost of information acquisition and, thus, revert to prior beliefs. We showed how varying an agent’s prior belief altered the resulting decisions and behaviour of agents, even those with equivalent utility functions. Importantly, the incorporation of prior beliefs into the decision-making framework allowed the separation of two key elements: The agent’s utility maximisation, and the contribution of the agent’s past beliefs. This separation allowed for a discussion on the decision-making process in a temporal sense, being able to refer to the previous decisions. This allows for investigation into the building of beliefs over time, elucidating resulting microeconomic foundations in terms of the underlying parameters.

It is worth pointing out some parallels with, and differences from, the frameworks of embodied intelligence and information-driven (guided) self-organisation, in which embodiment is seen as a fundamental principle for the organisation of biological and cognitive systems [63,64,65,66]. Similar to these approaches, we consider information-processing as a dynamic phenomenon and treat information as a quantity that flows between the agent and its environment. As a result, an adaptive decision-making behaviour emerges from these interactions under some constraints. Maximisation of potential information flows is often proposed as a universal utility for such emergent agent behaviour, guiding and shaping relevant decisions and actions within the perception-action loops [67,68,69,70]. Importantly, these studies incorporate a trade-off between minimising generic and task-independent information-processing costs and maximising expected utility, following the tradition of information bottleneck [71].

In our approach, we instead consider specific information acquisition costs incurred when the agents need to update their relevant beliefs in the presence of (Smithian) competition and market feedback. The adopted thermodynamic treatment of decision-making allows us to interpret relevant economic parameters in physical terms, e.g., agent’s decision temperature *T*, the strength of negative feedback ρ, and skewness of the resulting energy distribution γ. Interestingly, the decision temperature appears in our formalism as the Lagrange multiplier of the information cost incurred when switching posterior and prior beliefs (KL-divergence). The KL-divergence can be interpreted as the expected excess code-length that is needed if a non-optimal code that was optimal for the prior (outdated) belief is used instead of an optimal code based on the posterior (correct) belief. Thus, the decision temperature modulates the inference problem of determining the true distribution given new evidence, in a forward time direction [72]. Moreover, the thermodynamic time arrow (asymmetry) is maintained only when decision temperatures are non-zero.

We demonstrated the applicability of the method using actual Australian housing data, showing how the incorporation of prior knowledge can result in agents building on past beliefs. In particular, the agent focus can be shown to shift from utility maximisation to acting on previous knowledge. In other words, during the periods when the market has been performing well, the agents were shown to become overly optimistic based on the past performance.

The generality of the proposed approach makes it useful for incorporating any form of prior information on the agent’s choice set. Moreover, we have shown that the default QRSE is a special case of the proposed extension with uniform (i.e., uninformative) priors. Therefore, the proposed approach can be seen as an extension of QRSE, which accounts for prior agent beliefs based on information acquisition costs. As the QRSE framework continues to be expanded, the generalised model proposed here could become an important approach. Particularly, this would be useful whenever prior knowledge on agent decisions is known, as well as in multi-action cases when the IIA property of the general logit function is undesirable. Other relevant applications include scenarios with multiple time periods, allowing for a detailed temporal analysis and exploration of the cost of switching between equilibria (measured as an information acquisition cost from prior beliefs).

## Figures and Tables

**Figure 1 entropy-23-00669-f001:**
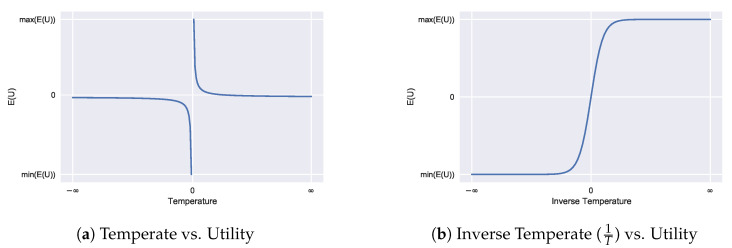
The effect of decision temperature *T* on the resulting expected payoffs (**a**). for the limits given by Equation (Equation 17). The inverse temperature $\frac{1}{T}$ (**b**) conveys the same information but may offer a more useful visualisation due to the continuity.

**Figure 2 entropy-23-00669-f002:**
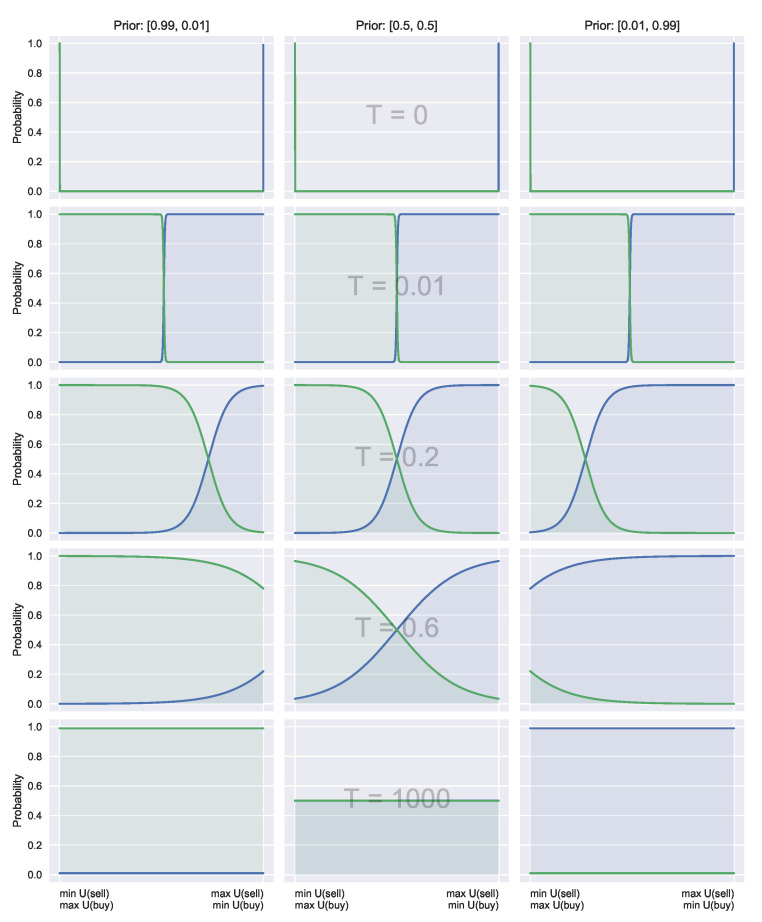
Decision Functions. All cases have equivalent utility functions. Each row has equivalent temperatures, showing how with matched parameters and utility, having an alternate prior can shift the decision-makers preference. Each column has different priors, given along the top of the first row to show how decision-makers decisions change based on their prior beliefs. On the left-hand side, preference is shifted towards the buying case. Likewise, on the right-hand side, preference is given to the selling case. The uniform case with equal preference is shown in the middle.

**Figure 3 entropy-23-00669-f003:**
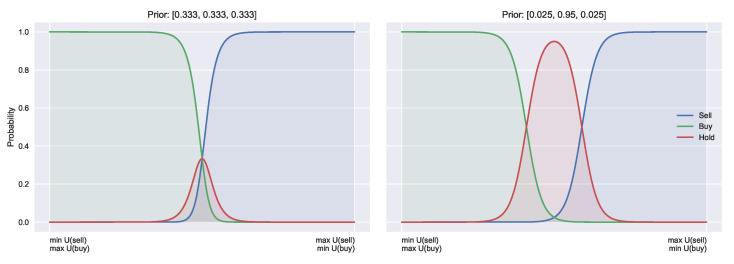
In the three-action case, the priors can introduce asymmetries by biasing the decision functions. This allows for separate indifferent points (**right**) vs. the uniform priors implying a single intersect (**left**).

**Figure 4 entropy-23-00669-f004:**
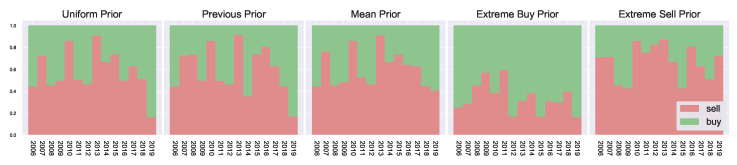
Resulting marginal probabilities f[a] for varying priors. Green represents f[buy], and red represents f[sell].

**Figure 5 entropy-23-00669-f005:**
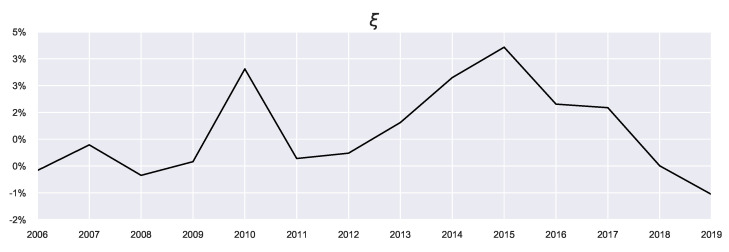
Real Average Returns.

**Figure 6 entropy-23-00669-f006:**
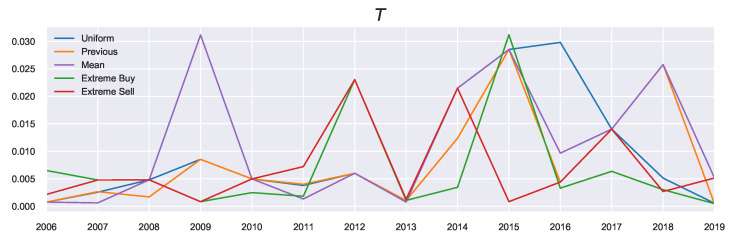
Decision Temperature.

**Figure 7 entropy-23-00669-f007:**
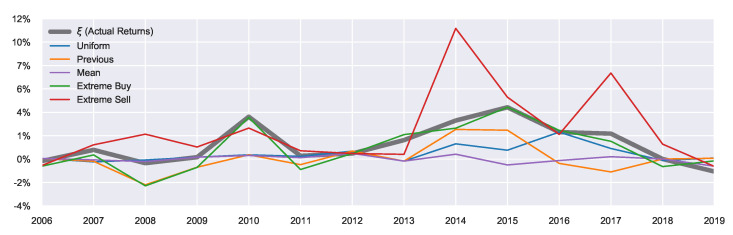
Agent Expectations vs. Actual Returns (in black).

**Figure 8 entropy-23-00669-f008:**
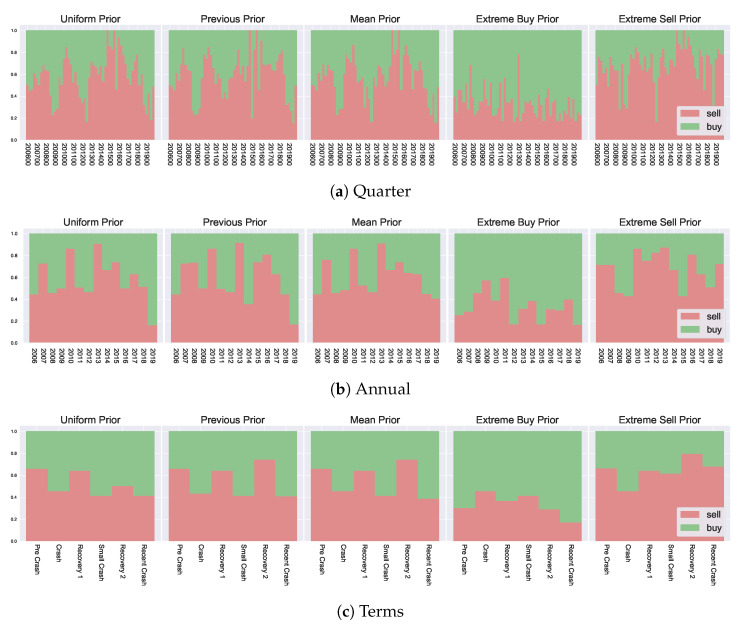
f[a] for varying granularities.

**Table 1 entropy-23-00669-t001:** Resuling likelihood and percentage of variability explained for each year, when compared to the actual underlying distribution (i.e., those given in Figure A2). Optimisation is done by minimising the negative log-likelihood between the resulting distributions and the actual distribution of returns.

	Uniform	Previous	Mean	Extreme Buy	Extreme Sell
**2006**	1082 (93%)	1082 (93%)	1082 (93%)	885 (59%)	1005 (74%)
**2007**	1089 (92%)	1089 (92%)	1090 (90%)	939 (68%)	1042 (83%)
**2008**	998 (95%)	905 (78%)	998 (95%)	998 (95%)	998 (95%)
**2009**	918 (96%)	918 (96%)	866 (88%)	880 (85%)	875 (85%)
**2010**	857 (95%)	857 (95%)	857 (95%)	740 (62%)	857 (95%)
**2011**	1045 (92%)	1044 (91%)	1047 (92%)	1045 (91%)	873 (62%)
**2012**	1067 (96%)	1067 (96%)	1067 (96%)	162 (6%)	142 (8%)
**2013**	1080 (90%)	1076 (90%)	1083 (90%)	983 (77%)	1075 (91%)
**2014**	938 (98%)	851 (74%)	938 (98%)	875 (71%)	938 (98%)
**2015**	860 (96%)	860 (96%)	860 (96%)	33 (10%)	808 (71%)
**2016**	873 (84%)	932 (95%)	908 (86%)	817 (70%)	932 (95%)
**2017**	916 (97%)	916 (97%)	916 (97%)	812 (76%)	916 (97%)
**2018**	989 (88%)	932 (85%)	933 (85%)	955 (82%)	998 (91%)
**2019**	1101 (92%)	1103 (92%)	1067 (94%)	1101 (92%)	952 (76%)

## Data Availability

The real-estate pricing data used in this work were made available under license for this study by SIRCA-CoreLogic (https://www.corelogic.com.au/industries/residential-real-estate).

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
