# Peer review of "A Maximum Entropy Model of Bounded Rational Decision-Making with Prior Beliefs and Market Feedback"

_entropy, 2021, doi:10.3390/e23060669_

Round 1

Reviewer 1 Report

Overall, I think this is an interesting and tightly written paper that deserves publication. It is clear that the literature around QRSE is evolving rapidly and I do think that the introduction of explicit priors into the model offers a valuable and novel extension. However, I find that the paper hand-waves across some details, uses some confusing language in other instances, and leaves a central conceptual question ambiguously open. I will discuss these in reverse order, reflecting their relative importance for preparing the paper for publication. All of my questions and concerns seem very fixable and believe the paper will be publishable in Entropyafter revision.

First, my conceptual concern(s). The papers central contribution is the introduction of prior beliefs captured by a distribution. Furthermore, this allows the authors to relate the informational costs of different actions to the Kullback-Leibler divergence between the prior distribution and the probability of taking difference actions conditional on the macro-state x. I think this is truly novel and a very worthwhile contribution to the model.

What is less clear is how the prior conceptualized this way differs from an agent’s expectations as given by \mu. Aren’t expectations also a form of prior? The prior p(x) seems to indicate uncertainty about the state of the world, x. Is p(a) not more of a predisposition of which action to take than a prior in the traditional sense? Where this ambiguity became most striking was in the demonstration exercise at the end using Australian housing prices. What exactly is the point of calibrating the model using different priors? Are the authors simply illustrating how agents with different priors act differently or are they implicitly making an inference about which priors (or action predilections) agents must have to explain the observed data? No judgement about fit is ever made, so the latter appears not to be the goal of the exercise, but then what is the goal?

My issue also goes back to the issues I have with the exposition of both the QRSE and thermodynamics of decision-making in section 3. Let’s start with the use of language. In 3.1.1, right before (1), the authors are missing the word “expected”: “agents are attempting to maximize their expected utility.” The objective function of the constrained maximization shown is clear expected utility. It would also be useful if the program included max and a formal indication that the functional of choice is f(a|x). Just below line 132 on p. 4, the authors write that “solving the problem corresponds to solving the following Lagrangian,” but it would be clearer if they described that this requires taking the first-order conditions and setting them equal to zero. As written, (3) is obviously not an equation that can be “solved” for (4). Most readers of Entropywould know this, of course, but it still seems like an unnecessarily misleading statement. The same issue arises in the description for how one gets from (12) to (13) and again for how the Lagrangian in (15) leads to the result in (16).

The deeper issue is that when these details are spelled out in a way that isn’t quite correct, the reader can lose confidence that the authors fully know what they are doing. A less trivial case of sits at the heart of the exposition of the QRSE model in section 3.1. What seems to be missing is explicit recognition that the constrained maximization given by (6) and (7) is the dual to (1) and (2), and that the dual implies a shift in perspective from agent to observer of the system (see Scharfenaker, 2020, in the Journal of Economic Dynamics and Control). Missing the shift in perspective of whose problem is being solved likely informs the confusion around the goal of the empirical exercise that I highlighted above.

The bottom line is that while I think the paper contains an interesting and novel idea, there are hints and indications that the authors have not fully worked out their own conceptual understanding of what they are doing. In pushing them to clarify key steps in their exposition, I hope their thinking will become clearer to the benefit of the reader.

Author Response

Thank you for the useful comments and feedback. Please see below for a point-by-point response to the comments raised. The updated manuscript has the changes indicated in red text. 

What is less clear is how the prior conceptualized this way differs from an agent’s expectations as given by \mu. Aren’t expectations also a form of prior? The prior p(x) seems to indicate uncertainty about the state of the world, x. Is p(a) not more of a predisposition of which action to take than a prior in the traditional sense?
We have tried to highlight this discussion in Section 4.3, as well as renaming the section, emphasizing the differences between mu and priors. Mu's can encapsulate priors (i.e. an adjusted mu can behave the same as a prior), but the important point is that mu is the result of an inference process, and is not a prior belief as information-processing must be performed.  p(a) can be seen as a predisposition of taking action before any information processing is performed which can be interpreted as a prior likelihood of taking an action before observing any utilities. We have highlighted the benefits of separating mu and p(a), as well as p(a) being applicable to alternate utility functions (as QRSE is not limited to the linear payoff with mu) in the new discussion in Section 4.3.

Where this ambiguity became most striking was in the demonstration exercise at the end using Australian housing prices. What exactly is the point of calibrating the model using different priors? Are the authors simply illustrating how agents with different priors act differently or are they implicitly making an inference about which priors (or action predilections) agents must have to explain the observed data? No judgement about fit is ever made, so the latter appears not to be the goal of the exercise, but then what is the goal?

We have added clarification to the end of Section 5s introductory preamble to motivate that we wish to explore and compare resulting dynamics from various priors, not necessarily arguing for a single best-fitting prior (or search for one). This shows the generality of the method for the incorporation of various priors.

We have also added an explanation that during crises, and periods exhibiting non-linear market dynamics, the macroeconomic conditions are likely to become more heterogeneous, and thus, non-uniform priors are expected to outperform uniform ones and we aimed to verify this conjecture (see Section 5).

In 3.1.1, right before (1), the authors are missing the word “expected”: “agents are attempting to maximize their expected utility.”

We have added the missing "expected".

It would also be useful if the program included max and a formal indication that the functional of choice is f(a|x).

We have added "max" to the various equations where this was missing (Equation 1 and Equation 14).

Just below line 132 on p. 4, the authors write that “solving the problem corresponds to solving the following Lagrangian,” but it would be clearer if they described that this requires taking the first-order conditions and setting them equal to zero. As written, (3) is obviously not an equation that can be “solved” for (4). Most readers of Entropywould know this, of course, but it still seems like an unnecessarily misleading statement. The same issue arises in the description for how one gets from (12) to (13) and again for how the Lagrangian in (15) leads to the result in (16).

We have reworded this to make the process clear throughout. Specifically, before Equations 4, 10, 13, and 16.

What seems to be missing is explicit recognition that the constrained maximization given by (6) and (7) is the dual to (1) and (2), and that the dual implies a shift in perspective from agent to observer of the system (see Scharfenaker, 2020, in the Journal of Economic Dynamics and Control). 

We have added discussion on the duality of maximising entropy subject to a constraint on expected utility, and maximising expected utility subject to a constraint on entropy. This has been added to the new Appendix A.1, along with a reference to the Sharfenaker JECD paper, and referenced after Equation 4.

We agree that Equation 6 and 7 follow the observer perspective. However, Equation 6 and 7 are not directly dual to Equation 1 and 2. The duality described in Sharfenaker is the one we have now also outlined in Appendix A.1. Hopefully we have made this clearer by the additional discussion added to Section 3.1.2, that there is a shift in perspective, and decisions can also be seen from the observer perspective based on the duality in Appendix A.1. 

Thank you again for the comments and feedback.

Reviewer 2 Report

This paper adds priors to the quantal response statistical equilibrium decision making model that has recently been proposed in this journal. The model is derived and illustrated  with an application to the Australian housing market. The rationale and derivation make sense to me. I have a few comments on the well-chosen example that I sketch below.

Some comments with line numbers:

All 'technical reports' cited should have a working paper series name, ideally also number.

107 This sentence "These developments show the usefulness of maximum entropy methods (specifically, QRSE) for inferring decisions from only macro-level economic data" could benefit from a footnote saying that maximum entropy methods were also applied earlier to economics if without a QRSE framework, notably by Yakovenko in several articles in the early 2000s, and by Scharfenaker and Semieniuk in 2017.

119 macroeconomic output - please define what that is; later 'macroeconomic property' and 'outcome' seem to be used to mean the same thing.

123 "QRSE infers the least biased actions" -- biased in what sense?

127 "entrance into a market tends to lower the profit rates"  -- unclear where suddenly profit rates come from in a general decision setting. Can you motivate this or give a more general notion of a negative feedback loop?

155 throughout literature. Insert 'the'

153 this sounds like the QRSE was first invented and then Boltzmann used it. Make sure to indicate the temporal order more clearly.

172 onwards. Please explain the example better, and ideally give one that's more relevant to Smithian competition. The current example does not help me understand IIA in the QRSE context.

Figure 2 x and y axes should have labels

The discussion in 306 onwards is interesting. It could be contrasted with Foley 2020, EPJST who who treats an agent as only ever selling or buying, revolving around an individual asset.

Section 5 - how is "Smithian competition" to be interpreted in this context?

I have to slightly more involved comments on the example, that I recommend the authors take some care to think about.

In the example, you speak of 'cost of acquisition' but it's not clear to me how that plays out in your example. Please give some intuition about how to understand this formulation in the context of economics which uses 'cost' usually associated with something measured in a monetary unit.

Table 1 - one slightly problematic argument for the pap seems to be that uniform priors perform no worse than the others. Then what's the advantage of having nonuniform priors from an Occam's razor perspective? Explain.

Author Response

Thank you for the useful comments and feedback. Please see below for a point-by-point response to the comments raised. The updated manuscript has the changes indicated in red text. 

All 'technical reports' cited should have a working paper series name, ideally also number.

These citations have now been fixed (Blackwell 2019, Omer 2018, and Weiner 2018).

107 This sentence "These developments show the usefulness of maximum entropy methods (specifically, QRSE) for inferring decisions from only macro-level economic data" could benefit from a footnote saying that maximum entropy methods were also applied earlier to economics if without a QRSE framework, notably by Yakovenko in several articles in the early 2000s, and by Scharfenaker and Semieniuk in 2017.

We have rephrased this sentence, as well as added the Scharfenaker and Semieniuk reference and several Yakovenko references to the background section (Paragraph 1 of Section 2). 

119 macroeconomic output - please define what that is; later 'macroeconomic property' and 'outcome' seem to be used to mean the same thing.

We have added an example definition when first using "macroeconomic outcome" in the introduction, as well as changed the wording to consistently use macroeconomic outcome throughout rather than property or output. 

123 "QRSE infers the least biased actions" -- biased in what sense?

We have updated this to say "least biased distribution of decisions", as well as added a footnote when we first mention least biased decisions to highlight the wording of Jaynes, i.e. least biased in an information-content setting.

127 "entrance into a market tends to lower the profit rates"  -- unclear where suddenly profit rates come from in a general decision setting. Can you motivate this or give a more general notion of a negative feedback loop?

We have rephrased this wording to motivate Smithian competition more generally before using the profit rate example.

155 throughout literature. Insert 'the'

Added.

153 this sounds like the QRSE was first invented and then Boltzmann used it. Make sure to indicate the temporal order more clearly.

We have reworded to make this clear that such developments arose independently. 

172 onwards. Please explain the example better, and ideally give one that's more relevant to Smithian competition. The current example does not help me understand IIA in the QRSE context.

We have reworded the example and added a specific sentence on portfolio allocation which will hopefully motivate IIA more in an economic sense.

Figure 2 x and y axes should have labels

Labels have now been added to Figure 2. 

The discussion in 306 onwards is interesting. It could be contrasted with Foley 2020, EPJST who who treats an agent as only ever selling or buying, revolving around an individual asset.

Thank you for this reference. This has been added to the background sentence, and a sentence added to the discussion relating to the transaction frequency discussed in Foley 2020. 

Section 5 - how is "Smithian competition" to be interpreted in this context?

We have added a sentence explaining Smithian competition in the context of housing markets to the end of Paragraph 1 in Section 5. 

In the example, you speak of 'cost of acquisition' but it's not clear to me how that plays out in your example. Please give some intuition about how to understand this formulation in the context of economics which uses 'cost' usually associated with something measured in a monetary unit.

We have reworded to specify cost of information acquisition wherever used, as well as added a specific example of what information costs mean in housing markets to Section 5.3.1 when discussing the decision temperature T.

Table 1 - one slightly problematic argument for the pap seems to be that uniform priors perform no worse than the others. Then what's the advantage of having nonuniform priors from an Occam's razor perspective? Explain.

We have added an explanation that during crises, and periods exhibiting non-linear market dynamics, the macroeconomic conditions are likely to become more heterogeneous, and thus, non-uniform priors are expected to outperform uniform ones and we aimed to verify this conjecture (see Section 5).

Thus, the macroeconomic complexity increases necessitating the non-uniformity of prior agent beliefs. In other times, the generalizability of the proposed model means that it is no more complex than the default QRSE, if we make the same assumption of uniform priors (see the explanation above Equation 16).

Thank you again for the comments and feedback.